# DEEP RANDOMIZED LEAST SQUARES VALUE ITERATION

## ABSTRACT

Exploration while learning representations is one of the main challenges Deep Reinforcement Learning (DRL) faces today. As the learned representation is dependant in the observed data, the exploration strategy has a crucial role. The popular DQN algorithm has improved significantly the capabilities of Reinforcement Learning (RL) algorithms to learn state representations from raw data, yet, it uses a naive exploration strategy which is statistically inefficient. The Randomized Least Squares Value Iteration (RLSVI) algorithm (Osband et al., 2016), on the other hand, explores and generalizes efficiently via linearly parameterized value functions. However, it is based on hand-designed state representation that requires prior engineering work for every environment. In this paper, we propose a Deep Learning adaptation for RLSVI. Rather than using hand-design state representation, we use a state representation that is being learned directly from the data by a DQN agent. As the representation is being optimized during the learning process, a key component for the suggested method is a likelihood matching mechanism, which adapts to the changing representations. We demonstrate the importance of the various properties of our algorithm on a toy problem and show that our method outperforms DQN in five Atari benchmarks, reaching competitive results with the Rainbow algorithm.

## 1 INTRODUCTION

In Reinforcement Learning (RL), an agent seeks to maximize the cumulative rewards obtained from interactions with an unknown environment (Sutton et al., 1998). Since the agent can learn only by its interactions with the environment, it faces the exploration-exploitation dilemma: Should it take actions that will maximize the rewards based on its current knowledge or instead take actions to potentially improve its knowledge in the hope of achieving better future performance. Thus, to find the optimal policy the agent needs to use an appropriate exploration strategy.

Classic RL algorithms were designed to face problems in the tabular settings where a table containing a value for each state-action pair can be stored in the computer's memory. For more general settings, where generalization is required, a common practice is to use hand-designed state representation (or state-action), upon which a function approximation can be learned to represent the value for each state and action. RL algorithms based on linear function approximation have demonstrated stability, data efficiency and enjoys convergence guarantees under mild assumptions (Tsitsiklis & Van Roy, 1997; Lagoudakis & Parr, 2003). They require that the desired learned function, e.g. Q-function, will be a linear combination of the state representation. This is, of course, a hard constraint as the representation is hand-designed, where the designer often does not know how the optimal value-function will look like. Furthermore, hand-designed representation is environment-specific and requires re-designing for every new environment.

The DQN algorithm (Mnih et al., 2015) has changed RL. Using Deep Neural Networks (DNN) as function approximators, the DQN algorithm enabled the learning of policies directly from raw high-dimensional data and led to unprecedented achievements over a wide variety of domains (Mnih et al., 2015). Over the years, many improvements to DQN were presented, suggesting more fitting network architectures (Wang et al., 2015), reducing overestimation (Van Hasselt et al., 2016; Anschel et al., 2017) or improving its data efficiency (Schaul et al., 2015). Despite its great success, DQN uses the overly simple $\epsilon$-greedy strategy for exploration. This strategy is one of the simplest exploration

strategies that currently exist. The agent takes random action with probability $\epsilon$ and takes the optimal action according to its current belief with probability $1 - \epsilon$. This strategy is commonly used despite its simplicity and proven inefficiency (Osband et al., 2016). The main shortcoming of $\epsilon$-greedy and similar strategies derives from the fact that they do not use observed data to improve exploration. To explore, it takes a completely random action, regardless of the experience obtained by the agent.

Thompson Sampling (TS) (Thompson, 1933), is one of the oldest heuristics to address the 'exploration/exploitation' trade-off in sequential decision-making problems. Its variations were proposed in RL (Wyatt, 1998; Strens, 2000) and various bandits settings (Chapelle & Li, 2011; Scott, 2010). For Multi-Armed Bandit (MAB) problems, TS is very effective both in theory (Agrawal & Goyal, 2012; 2013) and practice (Chapelle & Li, 2011). Intuitively, TS randomly takes actions according to the probability it believes to be optimal. In practice, a prior distribution is assumed over the model's parameters $p(w)$, and a posterior distribution $p(w|D)$ is computed using the Bayes theorem, where $D$ is the observed data. TS acts by sampling models from the posterior distribution, and plays the best action according to these samples.

Randomized Least Squares Value Iteration (Osband et al., 2016) is an RL algorithm which uses linear function approximation and is inspired by Thompson Sampling. It explores by sampling plausible $Q$-functions from uncertainty sets and selecting the action that optimizes the sampled models. This algorithm was proven to be efficient in tabular settings, with a bound on the expected regret that match the worst-case lower bound up to logarithmic factors. More importantly, it demonstrates efficiency even when generalization is required. Alas, as it assumes a linearly parametrized value function on a hand-designed state representation, the success of this algorithm crucially depends on the quality of the given state representation.

In this paper, we present a new DRL algorithm that combines the exploration mechanism of RLSVI with the representation learning mechanism of DQN; we call it the Deep Randomized Least Squares Value Iteration (DRLSVI) algorithm. We use standard DQN to learn state representation and explores by using the last layer's activations of DQN as state representation for RLSVI. To compensate for the constantly changing representation and the finite memory of DQN, we use a likelihood matching mechanism, which allows the transfer of information held by an old representation regarding past experience. We evaluate our method on a toy-problem – the Augmented Chain environment – for a qualitative evaluation of our method on a small MDP with a known optimal value function. Then, we compare our algorithm to the DQN and Rainbow algorithms on several Atari benchmarks. We show that it outperforms DQN both in learning speed and performance.

## 2 Related Work

**Thompson Sampling in Multi-Armed Bandit problems:** Thompson Sampling (TS) (Thompson, 1933), is one of the oldest heuristics to address the 'exploration/exploitation' trade-off in sequential decision-making problems. Chapelle & Li (2011) sparked much of the interest in Thompson Sampling in recent years. They rewrote the TS algorithm for Bernoulli bandit and showed impressive empirical results on synthetic and real data sets that demonstrate the effectiveness of the TS algorithm. Their results demonstrate why TS might be a better alternative to balance between exploration and exploitation in sequential decision-making problems than other popular alternatives like the Upper Confidence Bound algorithm (Auer et al., 2002). Agrawal & Goyal (2013) suggested a Thompson Sampling algorithm for the linear contextual bandit problem and supplied a high-probability regret bound for it. They use Bayesian Linear Regression (BLR) with Gaussian likelihood and Gaussian prior to design their version of Thompson Sampling algorithm. Riquelme et al. (2018) suggested performing a BLR on top of the representation of the last layer of a neural network. The predicted value $v_i$ for each action $a_i$ is given by $v_i = \beta_i^T z_x$, where $z_x$ is the output of the last hidden layer of the network for context $x$. While linear methods directly try to regress values $v$ on $x$, they independently trained a DNN to learn a representation $z$, and then used a BLR to regress $v$ on $z$, obtaining uncertainty estimates on the $\beta$'s, and making decisions accordingly via Thompson Sampling. Moreover, the network is only being used to find good representation – $z$. Since training the network and updating the BLR can be done independently, they train the network for a fixed number of iterations, then, perform a forward pass on all the training data to obtain the new $z_x$, which is then fed to the BLR. This procedure of evaluating the new representation for all the observed data is very costly, moreover, it requires infinite memory which obviously does not

scale. Zahavy & Mannor (2019) suggested matching the likelihood of the reward under old and new representation to avoid catastrophic forgetting when using such an algorithm with finite memory.

**Thompson Sampling in RL:** In the Reinforcement Learning settings, Strens (2000) suggested a method named "Posterior Sampling for Reinforcement Learning" (PSRL) which is an application of Thompson Sampling to Model-Based Reinforcement Learning. PSRL estimates the posterior distribution over MDPs. Each episode, the algorithm samples MDP from it and finds the optimal policy for this sampled MDP by dynamic programming. Recent work (Osband et al., 2013; Osband & Van Roy, 2017) have shown a theoretical analysis of PSRL that guarantees strong expected performance over a wide range of environments. The main problem with PSRL, like all model-based approaches, is that it may be applied to relatively small environments. The Randomized Least Squares Value Iteration (RLSVI) algorithm is an application of Thompson Sampling to Model-Free Reinforcement Learning. It explores by sampling plausible $Q$-functions from uncertainty sets and selecting the action that optimizes the sampled models.

**Thompson Sampling in DRL:** Various approaches have been suggested to extend the idea behind RLSVI to DRL. Bootstrapped DQN (Osband et al., 2017) uses an ensemble of Q-networks, each trained with slightly different data samples. To explore, Bootstrapped DQN randomly samples one of the networks and acts greedy with respect to it. Recently, Osband et al. (2018) extended this idea by supplying each member of the ensemble with a different prior. Fortunato et al. (2017) and Plappert et al. (2017) investigate a similar idea and propose to adaptively perturb the parameter-space, which can also be thought of as tracking approximate posterior over the network's parameters. O'Donoghue et al. (2017) proposed TS in combination with uncertainty Bellman equation, which connects the uncertainty at any time-step to the expected uncertainties at subsequent time-steps. Recently and most similar to our work, Azizzadenesheli et al. (2018) experimented with a Deep Learning extension to RLSVI. They changed the network architecture to exclude the last layer weights, optimized the hyper parameters and used double-DQN. In contrary, we don't change anything in the DQN agent. We use the representation learned by DQN to perform RLSVI, however, the network structure, loss and hyper-parameters are the same. Additionally, differently from our method, they don't compensate for the changing representation and solve BLR problem with the same arbitrary prior every time.

## 3 PRELIMINARIES

We consider the standard RL settings (Sutton et al., 1998), in which an environment with discrete time steps is modeled by a Markov Decision Process (MDP). An MDP is a tuple $< S, A, P, R, \gamma >$, where $S$ is a state space, $A$ a finite action space, $P : S \times A \to \Delta(S)$, is a transition kernel, and $R : S \times A \to \mathbb{R}$ a reward function. At each step the agent receives an observation $s_t \in S$ which represents the current physical state of the system, takes an action $a_t \in A$ which is applied to the environment, receives a scalar reward $r_t = r(s_t, a_t)$, and observes a new state $s_{t+1}$ which the environment transitions to. As mentioned above, the agent seeks an optimal policy $\pi^* : S \to \Delta(A)$, mapping an environment state to probabilities over the agent's executable actions. $\gamma \in (0, 1)$ is the discount factor – a scalar representing the trade-off between immediate and delayed reward. A brief survey of the DQN algorithm can be found in Appendix 1.

### 3.1 RANDOMIZED LEAST SQUARES VALUE ITERATION

The Randomized Least Squares Value Iteration (RLSVI) algorithm is a TS-inspired exploration strategy for Model-Free Reinforcement Learning. It combines TS-like exploration and linear function approximation, where the main novelty is in the manner in which it explores: Sampling value-functions rather than sampling actions. The Q-function is assumed to be in the form $Q(s, a) = \phi(s, a)^T w$, where $\phi(s, a)$ is a hand-designed state-action representation. RLSVI operates similar to other linear function algorithms and minimizes the Bellman equation by solving a regression problem – Bayesian Linear Regression. BLR obtains a posterior distribution over value-function instead of point estimates. To explore, RLSVI samples plausible value functions from the posterior distribution and acts the greedy action according to the sampled value-function. In the episodic settings where the representation is tabular, i.e., no generalization is needed, RLSVI guarantees near-optimal expected episodic regret. Finally, the main benefit of this algorithm is that it

displays impressive results even when generalization is required – despite the lack of theoretical guarantees. A pseudo-code can be found in Appendix 1.

# 4 THE DEEP RANDOMIZED LEAST SQUARES VALUE ITERATION ALGORITHM

In this paper, we propose to use RLSVI as the exploration mechanism for DQN. RLSVI capabilities are enhanced by using state representation that is learned directly from the data by a neural network rather than hand-designed one. As the neural network gradually improves its representation of the states, a likelihood matching mechanism is applied to transfer information from old to new representations.

## 4.1 LEARNING REPRESENTATION

A DQN agent is trained in the standard fashion, i.e., the same architecture, hyper-parameters and loss function as the original DQN. Two exceptions were made (1) The size of the last hidden layer is reduced to be $d = 64$. (2) The Experience Replay buffer is divided evenly between actions and transitions are stored in a round-robin fashion. I.e., whenever the buffer is full, a new transition $< s_t, a_t, r_t, s_{t+1} >$ is placed instead of the oldest transition with the same action $a_t$.

## 4.2 EXPLORATION

Exploration is performed using RLSVI on top of the last hidden layer of the target network. Given a state $s_t$, the activations of the last hidden layer of the target network applied to this state are denoted as $\phi(s_t) = \text{LastLayerActivations}(Q^{\theta_{\text{target}}}(s_t))$. Several changes to the original RLSVI algorithm were made: First, rather than solving different regression problem for every time step, a different regression problem is being solved for every action. As the last hidden layer's activations serves as state representation, the representation is time-homogeneous and shared among actions. The regression targets $y$ are DQN's targets which use the target network predictions. Another change is that a slightly different formulation of Bayesian Linear Regression than RLSVI is being used. Similar to RLSVI a Gaussian form for the likelihood is assumed: $Q(s, a) \sim N(w_a^T \phi(s), \sigma^2)$, however, like Riquelme et al. (2018) the noise parameter $\sigma^2$ is formulated as a random variable, which is distributed according to the Inverse-Gamma distribution. The prior for each regression problem is therefore in the form: $p(w, \sigma^2) = p(w|\sigma^2)p(\sigma^2), p(w|\sigma^2) \sim N(\mu_0, \sigma^2 \Sigma_0), p(\sigma^2) \sim InvGamma(a_0, b_0)$. For this prior and the Gaussian likelihood $Q(s, a) \sim N(w_a^T \phi(s), \sigma^2)$ it is known that the posterior distribution can be calculated analytically as follows:

$$\phi_n = \{\phi(s_1), ..., \phi(s_n)\} \in \mathbb{R}^{d \times n}, \quad Y_n = (y_1, ..., y_n)^T \in \mathbb{R}^n$$
$$\Sigma_n = (\phi_n \phi_n^T + \Sigma_0^{-1})^{-1}, \quad \mu_n = \Sigma_n(\phi_n Y_n + \Sigma_0^{-1}\mu_0),$$
$$a_n = a_0 + \frac{n}{2}, \quad b_n = b_0 + \frac{1}{2}(Y_n^T Y_n + \mu_0^T \Sigma_0^{-1} \mu_0 - \mu_n^T \Sigma_n^{-1} \mu_n), \tag{1}$$

Formulating $\sigma^2$ as a random variable allows adaptive exploration where the adaptation is derived directly from the observed data. Lastly, while RLSVI's choice for the prior's parameters is somewhat arbitrary, in our algorithm the prior has a central role which we'll discuss further on.

Since RLSVI requires a fixed representation and the target network's weights are fixed, we use the last layer activations of the target network, denoted $\phi(s)$, as state-representation. Every $T^{\text{target}}$ training time steps, the target network is updated with the weights of the $Q$-network. In these $T^{\text{target}}$ time steps the $Q$-network is changing due to the optimization performed by the DQN algorithm. Since the representation is changing, the posterior distribution that was approximated in the old representation can't be used. A posterior distribution based on the new representation needs to be approximated. Therefore, whenever the target network changes, new Bayesian linear regression problems are being solved using $N_{\text{BLR}}$ samples from the ER buffer. Since the ER buffer is finite, some experience-tuples were used to approximate the posterior in the old representation and are no longer available. Ignoring this lost experience can and will lead to degradation in performance derived by 'Catastrophic Forgetting' (Kirkpatrick et al., 2017). To compensate for the changing representation and the loss of old experience, we follow (Zahavy & Mannor, 2019) and match the

likelihood of the Q-function in the old and new representation. This approach assumes that the important information from old experiences is coded in the state representation.

---

**Algorithm 1** Deep RLSVI

---

**Input:** $s_0$ − Initial state, $Q^\theta(s,a)$, $Q^{\theta_{\text{target}}}(s,a)$, ER buffer,
   Prior: $\sigma_a^2 \sim \text{InvGamma}(a_{a,0}, b_{a,0})$, $p(w_{a,0}|\sigma_a^2) \sim N(\mu_{a,0}, \sigma^2 \Sigma_{a,0})$
   Define:   $\phi(s_t) \leftarrow \text{LastLayerActivation}(Q^{\theta_{\text{target}}})$,   $\psi(s_t) \leftarrow \text{LastLayerActivation}(Q^\theta)$
   **for** $t = 0, 1...$ **do**
     **if** $t \mod T_{sample}$ **then**
       **for** $a = 0, ..., |A|, j = 0, ..., |J|$ **do**
         Sample $\tilde{\sigma}_{a,j}^2 \sim \text{InvGamma}(\hat{a}_a, \hat{b}_a)$
         Sample $\tilde{w}_{a,j} \sim N(\hat{\mu}_a, \tilde{\sigma}_{a,j}^2 \hat{\Sigma}_a)$
       **end for**
     **end if**
     Sample $j_a \sim U\{1, 2, ..., |J|\} \forall a \in A$
     Act $a_t \in \arg\max_\alpha \tilde{w}_{\alpha,j_\alpha}^T \phi(s_t)$
     Observe $s_{t+1}, r_t$, Store Transition $< s_t, a_t, s_{t+1}, r_t >$
     Train DQN using sampled mini-batch
     **if** $t \mod T_{target} = 0$ **then**
       **for** $a = 0, ..., |A|$ **do**
         Construct Priors $\mu_0, \Sigma_0$ by likelihood matching (Equation 2)
         Sample $N_{\text{BLR}}$ transitions $< s_i, a, s_{i+1}, r_i >$ from ER buffer
         Solve Bayesian Linear Regression (Equation 1)
       **end for**
       $Q^{\theta_{target}}(s,a) \leftarrow Q^\theta(s,a)$
     **end if**
   **end for**

---

## 4.3 Constructing Priors by Likelihood matching

Recall that the likelihood of the Q function is $Q(s,a) \sim N(w_a^T \phi(s), \sigma^2)$, our best estimate for this likelihood is to plug-in the posterior approximation for $w_a$: $\hat{Q}(s,a) \sim (w_a^T \mu_n^\phi, \sigma^2 \phi^T(s) \Sigma_n^\phi \phi(s))$. The likelihood for the new representation, $\psi(s)$, is in the same form: $\hat{Q}(s,a) \sim (w_a^T \mu_n^\psi, \sigma^2 \psi^T(s) \Sigma_n^\psi \psi(s))$. Since the likelihood is Gaussian, to compensate for the changing representation, we will find moments that match the likelihood of the Q-function in the new representation to the old one and use them for our Gaussian prior belief.

**Expectation prior:** As DQN is trained to predict the $Q$-function, given the new last layer activations $\psi$, a good prior for $\mu_0$ in the new representation will be the last layer weights of the DQN (Levine et al., 2017).

**Covariance prior:** We use $N_{\text{SDP}}$ samples from the experience replay buffer. We evaluate both old and new representation $\{\phi(s_i), \psi(s_i)\}_{i=1}^{N_{\text{SDP}}}$. Our goal is to find a solution $\Sigma_0^\psi$ that will match the covariance of the likelihood in the new representation to the old one: $\psi(s_i)^T \Sigma_0^\psi \psi(s_i) = \phi(s_i)^T \Sigma_n^\phi \phi(s_i)$. Using the cyclic property of the trace, this is equivalent to finding $\text{Trace}(\psi(s_i)\psi(s_i)^T \Sigma_0^\psi) = S_i$, where $S_i = \phi(s_i)^T \Sigma_0^\phi \phi(s_i) = \text{Trace}(\phi(s_i)^T \Sigma_0^\phi \phi(s_i))$. We denote $\Psi_i = \psi(s_i)\psi(s_i)^T \in \mathbb{R}^{d \times d}$. Adding the constraint that $\Sigma_0^\psi$ should be Positive-Semi-Definite as it is a covariance matrix, we end up with the following Semi-Definite Program (SDP) (Vandenberghe & Boyd, 1996):

$$\min_{\Sigma_0^\psi} \sum_{i=0}^m ||\text{Trace}(\Psi_i \Sigma_0^\psi) - S_i||^2 \quad \text{s.t.} \quad \Sigma_0^\psi \succeq 0 \tag{2}$$

In practice, we solve this SDP by using CVXPY (Diamond & Boyd, 2016).

### 4.4 REDUCING COMPUTATION COMPLEXITY

**Approximate Sampling:** To perform Thompson Sampling one needs to sample the posterior distribution before every decision. This, regrettably, is computationally expensive and will slow down training significantly. To speed up learning, we sample $j$ weights for every action $\{\tilde{w}_{i,1}, ..., \tilde{w}_{i,j}\}$ every $T^{\text{Sample}}$ time steps. Then, every step we sample an index $i_a \in 1, .., j$ for every action, which is computationally cheap, and act greedy accordingly: $a_t = \arg\max_\alpha \tilde{w}_{\alpha,i_\alpha}^T \phi(s_t)$.

**Solving the SDP:** Another bottleneck our algorithm faces is solving the SDP. We refer the reader to Vandenberghe & Boyd (1996) for an excellent survey on the complexity of solving SDPs. As the running time of an SDP solver mainly depends on the dimension of the representation $d$, the number of samples being used $N_{\text{SDP}}$ and the desired accuracy $\epsilon$, we chose the last hidden layer size to be $d = 64$, used $N_{\text{SDP}} = 600$ for every SDP and set $\epsilon = 1e - 5$. The running time for solving a single SDP took us 10-50 seconds using Intel's Xeon CPU E5-2686 v4 2.30 GHz.

## 5 EXPERIMENTS

We conduct a series of experiments that highlight the different aspects of our method. We begin with a qualitative evaluation of our algorithm on a simple toy environment, then, move to report quantitative results on 5 different Atari games in the ALE.

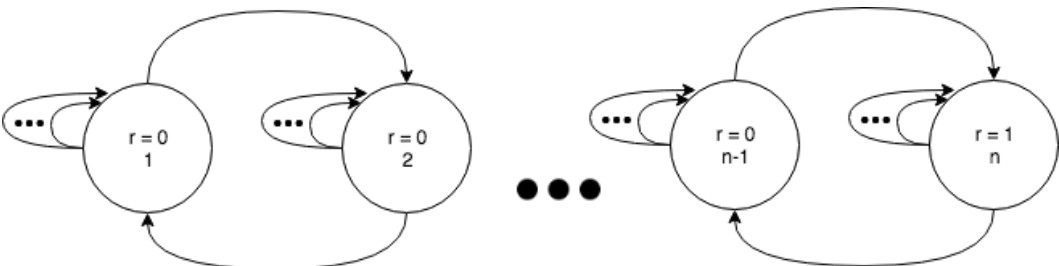

Figure 1: The Augmented Chain MDP

### 5.1 THE AUGMENTED CHAIN ENVIRONMENT

**Setup:** The chain environment includes a chain of states $S = 1, ..., n$. In each step, the agent can transition left or right. This standard-settings is augmented with additional $k$ actions which transitions the agent to the same state (self-loop). We name this variation "The Augmented Chain Environment". All states have zero rewards except for the far-right $n$-state which gives a reward of 1. Each episode is of length $H = n - 1$, and the agent will begin each episode at state 1. The raw state-representation is a one-hot vector. The $Q$-network is an MLP with 2 hidden layers. Results are averaged across 5 different runs. We report the cumulative episodic regret: Regret$(T) = \sum_{t=0}^{T}(v_0^*(s_0) - \sum_{h=0}^{H} r_{t,h})$. Here $T$ is the number of played episodes, $v_0^*(s_0)$ is the return of the optimal policy, and $r_{t,h}$ is the reward the agent received in episode $t$ at time step $h$. An illustration for the augmented chain environment can be found in figure 1. The hyper-parameters that are being used in the following experiments can be found in Appendix 2.

#### 5.1.1 EPSILON-GREEDY

We compared our algorithm to standard DQN where $\epsilon$-greedy serves as the exploration strategy. We experimented with various $\epsilon$ values, however, as the results for different $\epsilon$ values were similar, we display the result for a single $\epsilon$ value (Figure 2 (a)). We can see that $\epsilon$-greedy (red) achieves a linear regret in $T$ which is the lower bound for this type of problems, while our algorithm (blue) achieves much lower regret. These results demonstrate that $\epsilon$-greedy can be highly inefficient even in very simple scenarios.

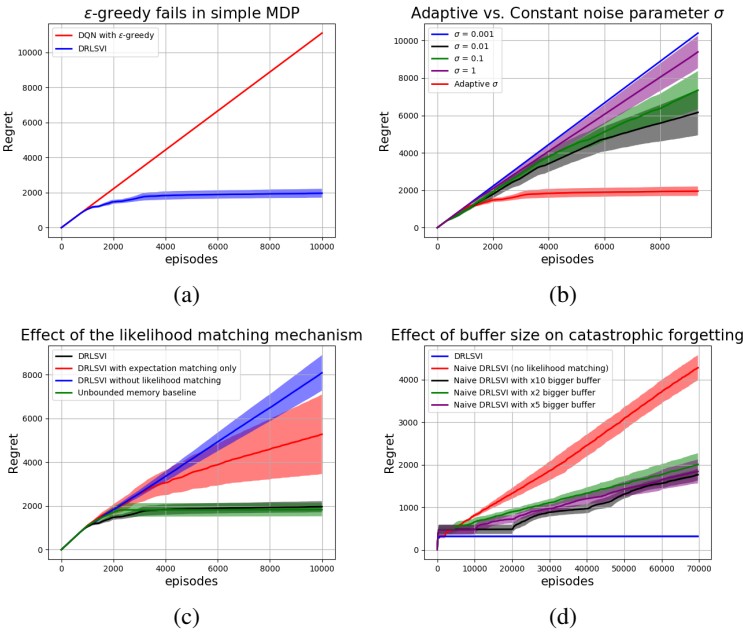

Figure 2: Experimental results in the "Augmented Chain Environment". (a) Comparison of $\epsilon$-greedy exploration with RLSVI exploration. (b) Effect of modeling the noise parameter as a random variable in comparison to different choices of a constant value. (c) Effect of the likelihood matching mechanism. (d) Effect of the buffer size.

### 5.1.2 ADAPTIVE SIGMA

In this experiment, we compared our algorithm with variants that do not model $\sigma^2$ as a random variable. We experimented with various constant $\sigma^2$ values. We can see that modeling $\sigma^2$ as a random variable (red) leads to lower regret compared to constant $\sigma^2$ variants (Figure 2 (b)). Note that choosing a small value for $\sigma^2$ (blue) results in near-deterministic posterior function. Therefore the results are very similar to the $\epsilon$-greedy variant. Intuitively, a deterministic posterior acts as a 0-greedy strategy. On the other hand, choosing a high value for $\sigma^2$ (purple) results in a very noisy sampling of the posterior approximation, therefore we get a policy which is relatively random concluding in a bad performance. Choosing $\sigma^2$ with the appropriate size for the given MDP (green, black) results in better performance, as indicated by the lower regret. However, as $\sigma^2$ is constant it doesn't adapt. We can see that the regret at the beginning of the learning is better even compared to the adaptive-version. However, as the uncertainty level decrease, the algorithm "over-explores" which results in inferior regret compared to the adaptive version.

### 5.1.3 LIKELIHOOD MATCHING

We compared our method with a variant that matches only the expectation, similar to (Levine et al., 2017), and a variant that does not match the likelihood at all, i.e., approximates the posterior with a fixed arbitrary prior. The version that does not match the likelihood at all is close to BDQN (Azizzadenesheli et al., 2018) and can be thought of as our implementation for it. Additionally, we report the results of a variant of the algorithm where the ER buffer is not bounded – this is possible due to the fact that the toy problem is very small and so choosing large enough buffer serves as infinite. Results are shown in Figure 2 (c). The superiority of our method (black) over the one-moment method (red) and no-prior at all (blue) support our claim that constructing priors by likelihood matching reduces the forgetting phenomenon. Additionally, the fact that the unbounded memory algorithm (green) doesn't demonstrate any degradation in performance confirms that this phenomenon is caused since the ER buffer is bounded.

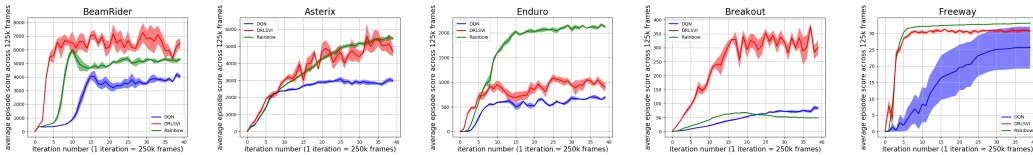

Figure 3: Learning curves of DQN(blue), DRLSVI(red), Rainbow(green) for the first 10M time steps, for 5 different Atari games

### 5.1.4 BUFFER SIZE

The previous experiment may suggest that catastrophic forgetting in DRLSVI can be avoided by simply increasing the buffer size. In the following experiment, We examine the simple Chain environment (no self-loop actions; $k = 0$), with the following modification: we replaced the meaning of the actions in half of the states, i.e., to move right in the odd states, the agent needs to take the opposite action from the even states. We compare our algorithm with variants that do not match the likelihood with different buffer sizes. Figure 2 (c) shows the performance of each of the algorithms in this setup. We can see that our algorithm (blue) doesn't suffer from degradation of performance. The other algorithms, that don't match the likelihood, all suffer from degradation, where the only difference is the time in which the degradation starts. These results demonstrate that without the likelihood matching mechanism, catastrophic forgetting will occur regardless of the buffer size. It is interesting to observe how catastrophic forgetting happens: When the buffer reaches a point where it doesn't contain experience of acting the non-optimal actions, a quick degradation occurs. Then, the algorithm initially succeeds to re-learn the optimal policy and the regret saturates. This phenomenon is getting increasingly aggravated until the regret becomes linear. These chain of events occurred in all the experiments without likelihood matching regardless of the buffer size.

### 5.2 THE ARCADE LEARNING ENVIRONMENT

We report the performance of our algorithm across 5 different Atari games. We trained our algorithm for 10 million time steps and followed the standard evaluation: Every 250k training time steps we evaluated the model for 125k time steps. Reported measurements are the average episode return during evaluation. For evaluation, we used the learned Q-network with $\epsilon$-greedy policy ($\epsilon = 0.001$), results are averaged across 5 different runs. We use the original DQN's hyper-parameters. Hyper-parameters that are only relevant for our method are summarised in Appendix 2. For comparison, we used the publically available learning curves for DQN[1] and Rainbow from the Dopamine framework (Castro et al., 2018). Rainbow (Hessel et al., 2018) is a complex agent comprised of multiple additions to the original DQN algorithm. The averaged scores for the three methods are presented in Figure 3. The evaluation suggests that our method explores in a much faster rate than DQN, and is competitive with the Rainbow algorithm that combines multiple improvements to the DQN.

**Note:** Azizzadenesheli et al. (2018) didn't supply standard evaluation metrics and reported results for a single run only. Additionally, they change the architecture of the Q-network to exclude last layer weights, so a direct comparison to our method is not feasible. We, therefore, didn't compare our results with theirs.

## 6 DISCUSSION

A Deep Learning adaptation to RLSVI was presented which learn the state representation directly from the data. We demonstrated the different properties of our method in experiments and showed the promise of our method. We hope to further reduce the complexity and running time of our algorithm in future work.

---

[1]In the publicly available results the authors use a different set of hyper-parameters than the original paper. We use the original paper's hyper-parameters. Notice that the results for DQN are generally the same for the new set of hyper-parameters, however, they may vary for a specific game

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

APPENDIX 1: ADDITIONAL PRELIMINARIES

DEEP Q-NETWORKS

The Deep Q-Networks (DQN) algorithm was the first algorithm that successfully combined Deep Learning architectures with Reinforcement Learning algorithms. It operates in the standard RL setting (Sutton et al., 1998) where the state space is high dimensional. It approximates the optimal Q-function using a Convolutional Neural Network (CNN). The algorithm maintains two DNNs, the Q-network with weights $\theta$ and the target network with weights $\theta_{\text{target}}$. The Q-Network gets a state $s$ as input and produce $|A|$ outputs, each one representing the Q-value of a different action $a$, $Q(s, a)$. The target network is an older version of the Q-network with fixed weights. The target network is used to constructs the targets $y$ that the Q-network is trained to predict. The targets $y$ are based on the Bellman equation. The algorithm uses Stochastic Gradient Descent (SGD) to update the network's weights, by minimizing the mean squared error of the Bellman equation defined as $\mathbb{E}[||Q_\theta(s_t, a_t) - y_t||^2$, where the target $y_t = r_t$ if $s_{t+1}$ is terminal, otherwise $y_t = r_t + \gamma \max_{a'} Q_{\theta_{\text{target}}}(s_{t+1}, a')$. The weights of the target network are set to $\theta$ every fixed number of time steps, $T^{\text{target}}$. The tuples $< s_t, a_t, r_t, s_{t+1} >$ that are used to optimize the network's weights are first collected into an Experience Replay (ER) buffer (Lin, 1993). When performing an optimization step, a mini-batch of samples are randomly selected from the buffer and are used to calculate the gradients. DQN is an off-policy algorithm which allows the agent to learn from experience collected by other means rather than its own experience. To explore the environment it applies the $\epsilon$-greedy strategy, i.e., with probability $\epsilon$ it takes random action and with probability $1 - \epsilon$ it takes the greedy action with respect to the current estimate of Q.

---

**Algorithm 2** Deep Q-Networks

---

**Input:** $Q^\theta(s, a), Q^{\theta_{target}}(s, a), \epsilon$, ER buffer
  $s_0 = \text{EnvironmentReset}()$
  **for** $t = 0, 1, ...$ **do**
    Sample $\epsilon_t \sim U(0, 1)$
    **if** $\epsilon_t < \epsilon$ **then**
      $a_t \sim U\{1, ..., |A|\}$
    **else**
      $a_t = \arg\max_\alpha Q^\theta(s, \alpha)$
    **end if**
    Act $a_t$
    Observe $s_{t+1}, r_t, d_t$ ($d_t = 1$ if $s_t$ is terminal)
    Store Transition $< s_t, a_t, s_{t+1}, r_t, d_t >$
    **if** $d_t = 1$ **then**
      $s_{t+1} = \text{EnvironmentReset}()$
    **end if**
    Sample $n$ transitions $< s_i, a_t, s_{i+1}, r_i, d_i >$ from ER buffer
    $y_i = r_i + (1 - d_t)\gamma \max_i Q^{\text{target}}(s_{t+1}, i)$
    $\theta \leftarrow \nabla_\theta ||Q(s_i, a_i) - y_i||^2$
    **if** $(t \mod T_{target}) = 0$ **then**
      $Q^{\theta_{target}}(s, a) \leftarrow Q^\theta(s, a)$
    **end if**
  **end for**

---

PSEUDO CODE FOR RLSVI

---

**Algorithm 3** Randomized Least Squares Value Iteration

---

**Input:** Feature Extractors: $\Phi_0, ..., \Phi_{H-1}$ ,
  Parameters: $\lambda > 0, \sigma > 0$
**Output:** $\tilde{\theta}_{i0}, ..., \tilde{\theta}_{iH-1}$
  Sample $\tilde{\theta}_{0,0}, ..., \tilde{\theta}_{0,H-1} \sim N(0, \sigma^2 \frac{1}{\lambda} I)$
  **for** $l = 0, 1...$ **do**
    Observe $s_0$
    **for** $h = H - 1, ..., 1, 0$ **do**
      Sample $a_{lh} \in \arg\max_\alpha (\Phi_h \tilde{\theta}_{lh})(s_{lh}, \alpha)$
      Act $a_{lh}$
      Observe $s_{l+1}, r_{lh}$
    **end for**
    Observe $r_{lH}$
    **for** $h = H - 1, ..., 1, 0$ **do**
      Generate regression problem $A \in \mathbb{R}^{l \times k}, b \in \mathbb{R}^l$
$$A \leftarrow \begin{bmatrix} \Phi_h(s_{0h}, a_{0h}) \\ \vdots \\ \Phi_h(s_{l-1,h}, a_{l-1,h}) \end{bmatrix}$$
$$b_i \leftarrow \begin{cases} r_{ih} + \max_\alpha (\Phi_{h+1}\tilde{\theta}_{l,h+1})(s_{i,h+1}, \alpha) & \text{for} \quad h < H - 1 \\ r_{ih} + r_{i,h+1} & \text{for} \quad h = H - 1 \end{cases}$$
      Bayesian Linear Regression
$$\overline{\theta}_{lh} \leftarrow \frac{1}{\sigma^2}(\frac{1}{\sigma^2}A^T A + \lambda I)^{-1} A^T b$$
$$\Sigma_{l,h} \leftarrow (\frac{1}{\sigma^2}A^T A + \lambda I)^{-1}$$
      Sample $\tilde{\theta}_{lh} \sim N(\overline{\theta}_{lh}, \Sigma_{l,h})$
    **end for**
  **end for**

---

## APPENDIX 2: HYPER-PARAMETERS

### AUGMENTED CHAIN EXPERIMENTS

| Hyper-Parameter | Value |
|---|---|
| mini-batch size | 32 |
| experience replay buffer size | 1000 |
| experience replay buffer size for oracle baseline | 1000000 |
| target network update frequency | 100 |
| discount factor | 0.99 |
| learning starts | 0 |
| $d$ - representation dimension | 20 |
| $T^{\text{sample}}$ - posterior sample frequency | 10 |
| $J$ - number of models | 5 |
| $N_{\text{BLR}}$ - Bayesian linear regression transitions | $1000/|a|$ |
| $N_{\text{SDP}}$ - likelihood matching transitions | 30 |
| $n$ - chain states | 10 |
| $k$ - self loop actions | 4 |

### ALE EXPERIMENTS

| Hyper-Parameter | Value |
|---|---|
| $d$ - representation dimension | 64 |
| $T^{\text{sample}}$ - posterior sample frequency | 1000 |
| $J$ - number of models | 5 |
| $N_{\text{BLR}}$ - Bayesian linear regression transitions | $1000000/|a|$ |
| $N_{\text{SDP}}$ - likelihood matching transitions | 600 |

