# OpenReview forum: "Deep Randomized Least Squares Value Iteration"
_ICLR.cc/2020/Conference — Reject_

### Official Review · AnonReviewer3 · 2019-10-17
**Official Blind Review #3**

**Rating:** 3

**Review:**

Deep Randomized Least Squares Value Iteration
=========================================================

This paper proposes a method for exploration via randomized value functions in Deep RL.
The algorithm performs a standard DQN update, but then acts according to an exploration policy sampled from a posterior approximation based on a last layer linear rule.
The authors show that this algorithm can perform well on a toy domain designed to require efficient exploration, together with some results on Atari games.


There are several things to like about this paper:
- The problem of efficient exploration in Deep RL is a pressing one, and there is no clearly effective method out there widely used.
- The proposed algorithm is interesting, and appears to have some reasonable properties. One nice thing is that it requires only relatively minor changes to the DQN algorithm.
- The general flow of the paper and structured progression is nice.
- The algorithm generally appears to bring superior exploration and outperform epsilon-greedy baseline.


However, there are some other places the work could be improved:
- I think that the name "Deep RLSVI" is a little imprecise... actually RLSVI could already be a "deep" algorithm as defined by the JMLR paper: http://jmlr.org/papers/volume20/18-339/18-339.pdf (Algorithm 4). I see that you mean this as an extension to the linear case for RLSVI... but I do think it would be better to call it something more explicit like "Last-layer RLSVI for DQN".
- Related to the above, the comparison to other similar methods for exploration via "randomized value functions" is not very comprehensive. I'm not sure what the pros/cons are of this method versus BootDQN or the very similar work from Azizzadenesheli?
- It would be good to compare these methods more explicitly, particularly on the domains designed specifically for testing exploration. To this end, I might suggest bsuite https://github.com/deepmind/bsuite and particularly the "deep sea" domains?
- Something feels a little off about the Atari results, particularly the curves for "rainbow"... these appear to be inconsistent with published results (look at Breakout).

Overall I think there is interesting material here, and I'd like to see more.
However, I do have some concerns about the treatment/comparison to related work and I think without this it's not ready for publication.

**Experience Assessment:**

I have published in this field for several years.

**Review Assessment: Checking Correctness Of Derivations And Theory:**

I assessed the sensibility of the derivations and theory.

**Review Assessment: Checking Correctness Of Experiments:**

I carefully checked the experiments.

**Review Assessment: Thoroughness In Paper Reading:**

I read the paper at least twice and used my best judgement in assessing the paper.

---

> ### Author Response · Authors · 2019-11-13
> **Reply to reviewer 3**
>
> - We will consider changing the algorithm’s name, thank you.
> - As you mentioned above, the goal of this paper is to demonstrate the benefit of replacing vanilla DQN’s exploration strategy to RLSVI. While other methods use an ensemble of networks (BootDQN) or change the network structure and loss (BDQN), we simply change the exploration strategy.
> - Thank you, we will consider it.
> - For the results of the baselines in the Atari section, we used the publicly available results in https://github.com/google/dopamine as supplied by DeepMind.

---

> > ### Comment · AnonReviewer3 · 2019-11-14
> > **Thank you**
> >
> > Thank you for your response.
> >
> > I'm not convinced by the argument that this "just changes the exploration strategy"... since it seems like this is linked to the other pieces of the algorithm in a way similar to the other algorithms...
> >
> > But I will have to consider you points more carefully... for now I am not convinced to upgrade to accept.
> >
> > Many thanks

---

> > > ### Author Response · Authors · 2019-11-15
> > > **Thank you!**
> > >
> > > In contrast to other algorithms like BDQN and BootDQN, we use the same network architecture, loss function and hyper-parameters of the original DQN. The main deviation from DQN is, therefore, the exploration strategy.
> > >
> > > Thank you!

---

### Official Review · AnonReviewer1 · 2019-10-22
**Official Blind Review #1**

**Rating:** 1

**Review:**

This paper introduces a deep learning-based adaptation for the RLVSI algorithm, where the agent uses the representation learned by the deep neural network-based RL agent (DQN). They use the last layer of DQN as a state representation for RLSVI.  In order to work with the changing representations of the deep agent, they propose a likelihood matching mechanism. The approach is applied to two tasks: a) A toy modified n-chain experiment and b) set of 5 Atari games. They show that their method outperforms the DQN with naive exploration.


This paper should be rejected because of the following reasons:

1) Lacking comparisons to Azizzadenesheli et al. (2018)
The authors acknowledge that their work resembles a lot to Azizzadenesheli et al. (2018), who also provide a deep extension of RLVSI. However, they do not provide any baselines or comparisons with this approach.  To me, this work is one particular form of the work by Azizzadenesheli et al. (2018), where instead of using Bayesian linear regression in the last layer, a specific parameterization of the prior and posterior families is used, that allows an analytical solution for the update (Eq 1).  They claim that they have lesser hyper-parameters but at the same time they introduce additional ones: N_{BLR}, T^{Sample}, etc. and it is not clear to me that if the new set of hyper-parameters are easier to tune than compared to Azizzadenesheli et al. (2018). They claim that it is not possible to compare with Azizzadenesheli et al. (2018)  but their code is available publicly.

2) Lacking comparisons in general
The results on Atari are compared with vanilla-DQN (with epsilon greedy). Instead of comparing the method on other works that also extend the RLVSI to deep nets (eg: [1], [2], [3], etc) they compare it with RAINBOW, a method that is not based RLVSI.

3) Unexplained design choices
A lot of design choices of the final algorithm are not explained, which makes me skeptical about the work. The main ones in question being:
The unique nature of the replay buffer: It is not clear why the experience replay buffer has the specific form, where each action has fixed memory, and a round-robin scheme is used to update the buffer.
) Non-standard experiments
It is not clear why the authors did not use the standard n-chain task but rather used the modified version.  Also, why did the authors only selected the set of only those 5 specific games is not addressed.


Suggestions
I will recommend the authors to address why their algorithm should be used instead of the others I have mentioned above. They can do it either by providing any theoretical or empirical arguments. They also should use a few of the standard experiments so that it gives the reader more insight into where their algorithms excel.


Things to improve the paper that did not impact the score:

Figure 3 is too small to read.
The section on Likelihood matching is not clear: in motivation and impact.


References:

[1] ] Ian Osband, Charles Blundell, Alexander Pritzel, and Benjamin Van Roy. Deep exploration via bootstrapped DQN. In Advances In Neural Information Processing Systems 29, pages 4026–4034, 2016.

[2] Ahmed Touati, Harsh Satija, Joshua Romoff, Joelle Pineau, and Pascal Vincent. Randomized value functions via multiplicative normalizing flows. arXiv preprint arXiv:1806.02315, 2018.

[3] Osband, Ian, John Aslanides, and Albin Cassirer. "Randomized prior functions for deep reinforcement learning." Advances in Neural Information Processing Systems. 2018.


**Experience Assessment:**

I have published one or two papers in this area.

**Review Assessment: Checking Correctness Of Derivations And Theory:**

N/A

**Review Assessment: Checking Correctness Of Experiments:**

I carefully checked the experiments.

**Review Assessment: Thoroughness In Paper Reading:**

I read the paper at least twice and used my best judgement in assessing the paper.

---

> ### Author Response · Authors · 2019-11-13
> **Reply to reviewer 1**
>
> 1. A direct comparison to BDQN is not straightforward since they do not have the last layer weights in their network architecture. We compared our algorithm to baselines which are similar in essence to BDQN like the ‘no prior’ baseline in section 5.1.3.
> 2. Thank you, we will consider adding these in a future version.
> 3. The reason for the round-robin mechanism is to allow the uncertainty to be derived from data quality rather than quantity. You are right that this is not highlighted in the text.

---

### Official Review · AnonReviewer2 · 2019-10-23
**Official Blind Review #2**

**Rating:** 1

**Review:**

The paper proposes to extend the popular linear-control algorithm, RLSVI, to utilize learned representations. This is done by adapting a work from bandit literature that utilizes BLR with representations that are learned via a DNN. The proposed solution is then compared to DQN with fixed epsilon as the exploration strategy in a chain MDP, and to the Rainbow agent and DQN in 5 selected Atari games to show sample-efficiency improvements.

The idea presented by the work is interesting, and utilizing a complete Bayesian Linear Regression framework (variable variance, as opposed to fixed variance in the prior) does sound appealing in terms of adaptability - as it is what the authors argue for, as being key to their proposal. But the work in the paper is not in an acceptable form due to the following key reasons: (1) the presentation of the ideas and the algorithm, "despite the lack of theoretical guarantees" is hard to understand, (2) the DQN baseline compared to (definitely in the Augmented Chain Environment, and possibly in the Atari games), are based on a fixed epsilon exploration strategy whereas DQN as proposed uses a epsilon-annealing strategy for exploration, (3) the baselines compared to are not comprehensive, (4) generally the paper is rather unpolished.


Following are main points of feedback, regarding which I would like to know the authors opinions/responses in the rebuttal if possible. After that, are some concrete questions to clarify the contents of the paper.

Feedback:
(1) the linear contextual bandit work the paper is built on is currently under review at ICLR 2020. While this should not be the reason for rejection, the idea of likelihood matching as motivated from that bandit work does not simply transfer to the RL case. For instance, based on the pseudocode in the paper (and the provided explanation), after likelihood matching is done, the new BLR updates still seem to use the old representations and targets (as the target net update is done after the loop). This is possibly a typo, or something deeper is left unexplained here.
(2) While it is true that the BDQN work does not utilize the last layer of their weights, and is built on the DDQN algorithm, I think they still are a reasonable baseline as the key idea distinguishing the two is variable variance prior vs. a fixed variance prior. While I understand having all the baseline experiments from the exploration in deep RL literature is hard as it is quite vast - Bootstrap DQN, UBE, Bootstrap Prior - I think the closest baseline has to be BDQN to your proposal and therefore is a natural competitor.
(3) DDQN has been shown to be reducing the overestimation bias prevalent in DQN and therefore was the framework BDQN was built on. Why do you choose to use DQN instead of DDQN? A discussion regarding this seems a natural part of the paper.
(4) Are the benefits of adaptive sigma present if the base algorithm is changed to DDQN? I think this is an important point of discussion/analysis. It does not have to outperform DDQN, but even a comparative study empirically would be an insightful and comprehensive contribution.
(5) I assume that the likelihood utilizes the inverse of the covariance as opposed to the covariance -- something that seems amiss in the current write-up.
(6) Isn't S_i in Section 4.3 already a scalar? Why is there a trace of a scalar?
(7) Section 5.1.3 -- the version that does not match likelihood is not BDQN as it is built on DDQN -- presumable significant regret differences.

Questions:
(1) DQN as proposed is with annealed e-greedy. Are the experiments in Augmented Chain Environment utilizing this or a fixed e-greedy strategy?
(2) What exactly is the connection between catastrophic forgetting and likelihood matching in the policy improvement context? Why should an improved policy match the likelihood of features as learned by an older policy?
(3) Why choose to sample a set of value functions instead of sampling every timestep? Only because sampling is expensive or does it provide any stability?
(4) Why did you pick these 5 Atari games?


Given the algorithm is a particular design choice, and the argument for its utility is empirical, I definitely think the design choice needs to be discussed more thoroughly, and the manuscript currently does not do that. While empirical experiments in the Atari suite can be hard ask depending on the availability of computational resources, I think the work algorithmic choices made here are left undiscussed and the empirical results aren't really convincing. Further, the presentation is rather imprecise and error-ridden. Therefore, I do not think the work can be accepted.

There are many imprecise statements and typos in the paper, which are listed here to aid the future versions:
(1) Please review your psuedocode based on comment (1) in Feedback.
(2) The content in the Introduction does not make a note of many algorithms proposed for exploration in Deep RL -- UBE [1], Bootstrap DQN [2], Randomized prior for Bootstrap DQN [3], Parameter noise[4].
(3) Section 2, para 2 -- Recent work (Osband --> Recent works (Osband
(4) Section 2, para 3 -- acts greedy --> acts greedily
(5) Section 3, para 1 -- gamma in (0,1) --> [0,1]
(6) Section 3, para 1 -- survey --> review
(7) Section 3.1, para 1 -- acts the greedy action according to the --> acts greedily wrt
(8) Section 4.1, buffer consists of gamma/termination flag
(9) Section 4.2, para 1 -- First, rather than solving .. -- so did RLSVI. I guess you mean the regression problem is not solved every tilmestep.
(10) Section 4.2 last sentence -- please expand why that assumption is good?
(11) opening inverted commas everywhere.
(12) Section 5.1.4, last sentence -- grammar.
(13) Footnote 1 and content in Section 5.2 "we used publicly available learning curves" are contradictory.


[1] O'Donoghue, Brendan, et al. "The uncertainty bellman equation and exploration." arXiv preprint arXiv:1709.05380 (2017).
[2] Osband, Ian, et al. "Deep exploration via bootstrapped DQN." Advances in neural information processing systems. 2016.
[3] Osband, Ian, John Aslanides, and Albin Cassirer. "Randomized prior functions for deep reinforcement learning." Advances in Neural Information Processing Systems. 2018.
[4] Plappert, Matthias, et al. "Parameter space noise for exploration." arXiv preprint arXiv:1706.01905 (2017).



**Experience Assessment:**

I have published one or two papers in this area.

**Review Assessment: Checking Correctness Of Derivations And Theory:**

I carefully checked the derivations and theory.

**Review Assessment: Checking Correctness Of Experiments:**

I carefully checked the experiments.

**Review Assessment: Thoroughness In Paper Reading:**

I read the paper thoroughly.

---

> ### Author Response · Authors · 2019-11-13
> **Reply to reviewer 2**
>
> Statements:
> 1. After the likelihood matching procedure, we only use the new representations. We will make it clearer in future versions.
> 2. First, we would like to comment that the key difference between BDQN and DRLSVI is the likelihood-matching mechanism. While a direct comparison to BDQN is not included in our paper, we demonstrate that our main deviations from BDQN are essentials in sections 5.1.2 and 5.1.3. In the ALE section, we did not compare our results to theirs as they do not supply standard evaluation metrics.
> 3. Thank you for raising this point. While in general, it may be beneficial to combine TS and DDQN (as in BDQN), our paper focus is on the effect of TS exploration on vanilla DQN - without any other additions.
> 4. In this paper, we focused on a DQN version. Thank you for your suggestion, we may consider this comparison in future versions.
> 5. Thank you, we will improve the notations.
> 6. Yes, you are right. It is written for clarity.
> 7. We didn’t claim it to be BDQN only mentioned it's similar to it. However, while BDQN may end up with lower regret, this may be attributed to the use of DDQN rather than to the use of RLSVI-like exploration which was the focus of our paper.
>
> Questions:
> 1. The experiments in the Augmented Chain Environment use fixed e-greedy strategy.
> 2. The likelihood matching is used to prevent forgetting of experience that is no longer available. An improved policy would benefit from such experience as demonstrated in section 5.1.4. Furthermore, the alternative approach is to use an arbitrary prior which is obviously less beneficial.
> 3. Because sampling is computationally expensive.
> 4. We used the taxonomy from [1] and chose easy and hard exploration games.
>
> Thank you for your thorough review and remarks.
>
> [1] Bellamere, Marc, et al. Unifying Count-Based Exploration and Intrinsic Motivation.

---

> > ### Comment · AnonReviewer2 · 2019-11-15
> > **Interesting idea, but not acceptable in current form**
> >
> > Thank you for your response.
> >
> > I think even if DQN is a baseline, it should be utilized with epsilon-annealing exploration, instead of fixed -- as it was proposed. Further, the choices made, for example - proposal built on top of DQN (instead of DDQN), fixed sampling etc. -- need to be thoroughly explored. The work currently is possibly a nice idea, with very limited empirical evidence and the presentation is rather confusing. Exploring the choices made, and comparing to baselines like - Bootstrap DQN, Randomized priors and BDQN -- seems imperative.
> >
> > Therefore, I'm changing my score to a reject, and I hope the future versions are more comprehensive and clear. Thank you.

---

### Decision · Program_Chairs · 2019-12-19

**Decision:**

Reject

**Comment:**

This paper combines DQN and Randomized value functions for exploration.

All the reviewers agreed the paper is not yet ready for publication. The experiments lack appropriate baselines and thus it is unclear how this new approach improves exploration in Deep RL. The reviewers also found some of the algorithmic design decisions unintuitive and unexplained. The authors main response was the objective was to improve and compare against vanilla DQN. This could be a valid goal, but it requires clear motivation (perhaps the focus is on simply algorithms that are commonly used in applications or something). Even then comparisons with other methods would be of interest to quantify how much the base algorithm is improved, and to justify empirically all the design decisions that went into building such an improvement (performance vs complexity of implementation etc).

The reviewers gave nice suggestions for improvements.  This is a good area of study: keep going!